# Publicly available data reveals association between asthma hospitalizations and unconventional natural gas development in Pennsylvania

Anna Bushong[1,2,¤], Thomas McKeon[3,4], Mary Regina Boland[3,5], Jeffrey Field[2,3]*

**1** Biology Program, Centre College, Danville, KY, United States of America, **2** Department of Systems Pharmacology and Translational Therapeutics, University of Pennsylvania School of Medicine, Philadelphia, PA, United States of America, **3** Center of Excellence in Environmental Toxicology, University of Pennsylvania, Philadelphia, PA, United States of America, **4** Department of Geography and Urban Studies, Temple University, Philadelphia, PA, United States of America, **5** Department of Biostatistics, Epidemiology and Informatics, Perelman School of Medicine

¤ Current address: Purdue University, Department of Forestry and Natural Resources, West Lafayette, IN, United States of America

* jfield@upenn.edu

**Data Availability Statement:** Primary data from this study and code are available at GitHub: https://github.com/bolandlab/Bushong_HydraulicFracturingPaper_PlosOne_2022

## Abstract

Since the early 2000s, unconventional natural gas development (UNGD) has rapidly grown throughout Pennsylvania. UNGD extracts natural gas using a relatively new method known as hydraulic fracturing (HF). Here we addressed the association of HF with asthma Hospitalization Admission Rates (HAR) using publicly available data. Using public county-level data from the Pennsylvania Department of Health (PA-DOH) and the Pennsylvania Department of Environmental Protection for the years 2001–2014, we constructed regression models to study the previously observed association between asthma exacerbation and HF. After considering multicollinearity, county-level demographics and area-level covariables were included to account for known asthma risk factors. We found a significant positive association between the asthma HAR and annual well density for all the counties in the state (3% increase in HAR attributable to HF, $p<0.001$). For a sensitivity analysis, we excluded urban counties (urban counties have higher asthma exacerbations) and focused on rural counties for the years 2005–2014 and found a significant association (3.31% increase in HAR attributable to HF in rural counties, $p<0.001$). An even stronger association was found between asthma hospitalization admission rates (HAR) and $PM_{2.5}$ levels (7.52% increase in HAR attributable to $PM_{2.5}$, $p<0.001$). As expected, asthma HAR was significantly higher in urban compared to rural counties and showed a significant racial disparity. We conclude that publicly available data at the county-level supports an association between an increase in asthma HAR and UNGD in rural counties in Pennsylvania.

**Funding:** This study was supported in part by the National Institute of Health R25ES021649 and P30ES013508 to JF. The funders had no role in study design, data collection and analysis, decision to publish, or preparation of the manuscript.

**Competing interests:** The authors declare that they have no known conflicts of interests that could appear to influence the work in this manuscript.

## Introduction

Petroleum and natural gas production has steadily increased in the United States over the past two decades [1]. As of 2021, the United States is a world leader in the production and export of oil and gas [2]. The United States became a net total energy exporter in 2019, which had not occurred since 1952. The increase in natural gas production played a vital role in the transition of the US from net total energy importer to an exporter. The increase in gas production is attributable to technological advancements called unconventional natural gas development (UNGD). UNGD combines horizontal drilling with a well stimulation technique called hydraulic fracturing (HF), or fracking. UNGD horizontal drilling enables well operators to follow thin rock layers with deposits of natural gas or oil that reside 1 mile or further below the Earth's surface, which were previously inaccessible using conventional methods and then stimulate production by HF [3]. Since 2008, UNGD has dramatically increased in Pennsylvania because it became economical for producers to extract gas from new regions of the Marcellus Shale, which lies underneath two-thirds of the state (Fig 1). Since the commercial introduction of UNGD in 2004, nearly 12,000 UNGD wells have been drilled in Pennsylvania [4]. Hydraulic fracturing rapidly changed Pennsylvania by bringing heavy industry to sparsely populated rural areas.

There is broad public concern about the possible adverse health risks to people living near UNGD because, even though a majority of HF wells are in rural areas, more than 10% of Pennsylvanians live within one mile of an active HF well [5]. Prior epidemiology studies found associations between fracking and several adverse medical conditions including heart failure, low birth weight, preterm birth and asthma [6–9].

Asthma is a chronic airway disease that affects over 1 million Pennsylvanians making it a significant public health concern [10]. Severe asthma exacerbation is an appropriate health metric to consider when investigating the community impact of UNGD because it can be triggered by environmental factors, such as poor air quality and stress [11–13]. UNGD can release environmental toxicants that could exacerbate asthma, such as volatile organic compounds

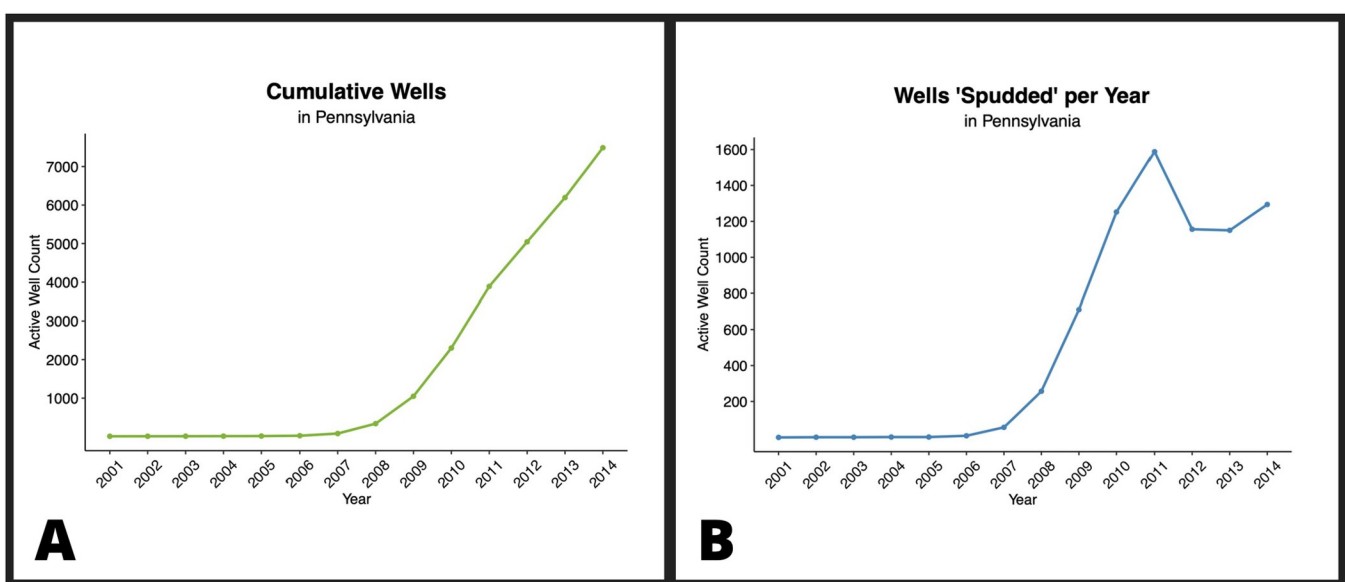

**Fig 1. Line graphs of UNGD in Pennsylvania over time.** Panel A shows the number of active wells that have been drilled over time as determined through each well's corresponding spud date (defined as the date well drilling commenced or date the largest pipe casing for the well was installed). Panel B shows the number of active wells that have been drilled each calendar year as determined through each well's corresponding spud date.

(VOCs), nitrogen dioxide ($NO_x$), diesel exhaust, silica dust, and particulate matter [14]. Additionally, unlike other diseases such as cancer, there can be a short latency between exposure to an environmental trigger associated with UNGD and a severe asthma exacerbation [15].

Prior studies on the effects of HF on health relied on Protected Health Information (PHI) from private sources that are not publicly available, such as hospital or insurance records. For example, an asthma study relied on private patient health records that included residential addresses and individualized demographic information to geocode patients [7]. Publicly available data provides the opportunity to investigate the influence of UNGD on asthma exacerbation while circumnavigating concerns regarding individual patient reidentification. In public data bases, severe asthma exacerbations are recorded as hospital admissions and can be quantified as asthma hospital admission rates (asthma HAR).

Here, we investigated the association between UNGD and asthma hospitalization admission rates (HAR), using HAR as a proxy for severe asthma exacerbation. We developed two multiple linear regression models using publicly available data at the county-level for the state of Pennsylvania, while accounting for relevant environmental and socioeconomic covariables. Both models found an association between UNGD and asthma HAR.

## Materials and methods

### Approach and sourcing of data

Asthma is reported by the Pennsylvania Department of Health publicly as hospitalization admission rates. Asthma HAR are reported by 62 of the 67 counties each year. Other publicly available data for variables of interest were sourced from state and federal government agencies. All data sourced for statistical analysis was organized into a singular spreadsheet for importation and analysis in R studio. We used the years 2005 to 2014 because they cover the periods before and after fracking began in PA. Also, after 2014 changes in the medical coding system made comparisons of hospitalizations difficult with earlier years.

**Asthma hospitalization admissions by county.** The Pennsylvania Department of Health (PA-DOH) compiles and releases data on various health metrics, including age-adjusted asthma HAR sourced through the Enterprise Data Dissemination Informatics Exchange (EDDIE), an interactive health statistics dissemination web tool. These data were provided by the Division of Health Informatics, Pennsylvania Department of Health. The Department specifically disclaims responsibility for any analyses, interpretations, or conclusions [16].

**Unconventional oil & gas wells by county.** Data for active UNGD wells in the state were obtained from the publicly available operator well inventory maintained by Pennsylvania Department of Environmental Protection (PA-DEP) [4]. Active UNGD wells that did not have a specified spud date (defined as the date well drilling commenced or date the largest pipe casing for the well was installed) (771 wells) were excluded from analysis as this date is necessary to determine if drilling or construction had commenced for a specific well. Active UNGD wells from the years 2001–2014 (7489 wells) were then sorted by date and matched to their county to determine the number of wells spudded each year for every county. We calculated the well density for active UNGD wells spudded during a calendar year in each county using the area of each county provided by the US Census Bureau through TIGERweb [17].

**Geographic data for counties.** We used spatial data to generate county-level maps of Pennsylvania for visual analysis. Spatial data for county boundaries were sourced from the Pennsylvania Department of Transportation provided through Pennsylvania Spatial Data Access (PASDA) and joined by county name to visualize relevant data of interest, including asthma hospitalization rates, population density, and fine particulate ($PM_{2.5}$) pollution [2]. Unconventional wells were mapped as point data utilizing each well's latitude and longitude

and layering over a county-level map of Pennsylvania. All mappings were done in QGIS 3.16.1-Hannover and R version 4.0.3, utilizing the coordinate reference system NAD83 and the ggmap, rgdal, sf, and sp packages [18–21]. The full reproducible code is available in GitHub.

**Demographic covariates.** Demographic covariables were considered during statistical model building to control for potential confounding effects when examining the relationship between the annual well density of unconventional natural gas wells and asthma hospitalization. All demographic data were publicly sourced at the county-level from the US Census Bureau through the Small Area Health Insurance Estimates (SAHIE), the American Community Survey (ACS) five-year estimates for 2005–2009 and 2010–2014, and the 2010 Census Urban and Rural Classifications. We extracted data on the percentage of people uninsured under the age of 65 from the SAHIE. The ACS provided estimates of county-level median household income, percentage of white population and percentage of people over 25 years of age with a high school diploma or equivalent. The latter two demographic variables were computed for each county in PA using the population estimates provided by each ACS five-year survey. The 2010 Census Urban and Rural Classifications provided the percent of people living in urban areas at the county-level. For our analyses, we also classified PA counties as rural or urban, based on population density being either below or above the population density for the whole state (Fig 2A). The assignment of counties as urban or rural is based on the classification presented in the PA-DOH 2012 Asthma Burden Report, which matches the classifications of The Center for Rural Pennsylvania (CRP) [2, 22]

**Area-level covariates.** Relevant environmental covariables were also considered during statistical model building to control for confounding effects, which were $PM_{2.5}$ pollution and smoking prevalence at the county-level. We extracted modeled and monitored data for annual average ambient concentration of $PM_{2.5}$ in $\mu g/m^3$ from the Centers of Disease Control and Prevention (CDC). The percentage of current smokers was sourced from the PA-DOH through the Behavioral Risk Factor Surveillance System (BRFSS).

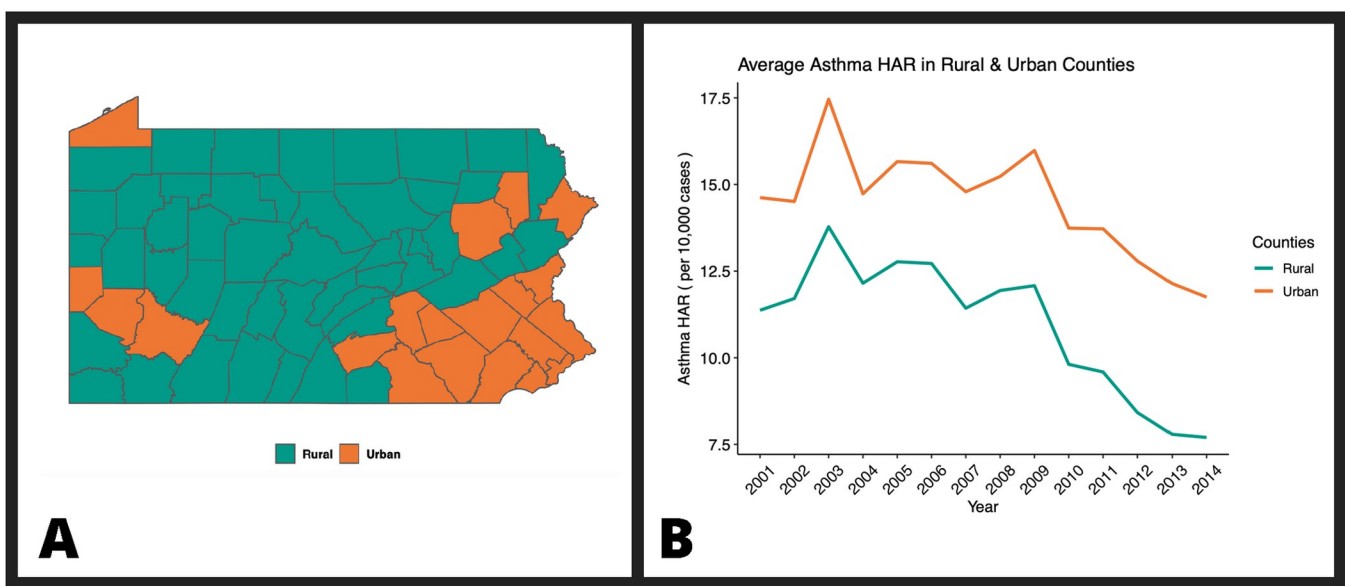

**Fig 2. Comparison of annual average asthma hospitalization admission rates (HAR) in rural and urban counties of Pennsylvania.** Panel A shows a map of Pennsylvania with counties designated as urban or rural as distinguished by the PA-DOH. The orange counties are those designated as urban, while the green represent counties that are rural. Panel B shows a line graph of the average asthma HAR per year from 2001–2014 for counties designated as rural and urban by the PA-DOH. The orange line represents the average asthma HAR per year for urban counties, while the green represents rural counties.

## Statistical analysis

Data analysis was performed, and graphs created using R version 4.0.3 and the ggplot2 and dplyr packages [23, 24]. Reproducible code is provided in the supplementary materials with additional packages [18–21, 25–27]. We used data for county-level asthma HAR as the response variable for this study after performing a natural log-transformation to help normalize its right-skewed distribution. However, an age-adjusted hospitalization admission rate for a county was not displayed by the PA-DOH if its raw count was below 20, regardless of county population size, so these missing data points were not included for analysis. After computing the annual well density on the county-level, we used this metric as our main explanatory variable for this study.

Multicollinearity between covariables was investigated to minimize redundancy in our final models. With some subjectivity in terms of what is appropriate for the upper limit of correlation between covariables, we choose a conservative threshold of approximately 50% correlation as the threshold for multicollinearity permitted. After performing correlation tests, we determined the following: (a) median household income could not be included in the same model as the percentage of people percentage of people over 25 with a high school diploma or equivalent, or percentage of current smokers, (b) percentage of people uninsured under 65 could not be included in the same model as percentage of people over 25 with a high school diploma or equivalent, and (c) percentage of population in urban areas could not be included in the same model as percentage of white population.

We utilized multiple linear regression models with fixed-year effects to investigate the relationship between the annual density of HF wells and asthma HAR. To meet the objects of our study, we built two separate regression models. The first regression model was built to investigate the relationship between asthma HAR and HF across all the counties, including both urban and rural counties. The second regression model was built to optimize the regression model for rural counties.

When building these models, we considered covariables with a socioeconomic or biological plausibility to exacerbate asthma (specifically, (a) annual average ambient air concentration pf $PM_{2.5}$, (b) percentage of people uninsured under 65, (c) median household income, percentage of white population, (d) percentage of population over 25 with at least a high school diploma or equivalent, (e) percentage of population that are current smokers, (f) percentage of population living in urban areas) [7, 14, 28]. We utilized a stepwise regression tool that is AIC-based and uses backward elimination for initial investigation of the input variables' influence on the model. To adequately consider issues of multicollinearity, models were then manually evaluated with consideration for the models' residual standard error, adjusted R-squared, p-value, and covariables' significance to select our model with annual well density as the main explanatory variable. Due to the response variable being natural log-transformed, the partial slopes for these regression models were appropriately back-transformed by exponentiating the slope, subtracting one, and multiplying by 100. This enabled the partial slopes to be interpreted as a percent increase in asthma HAR per unit increase for that explanatory variable. 95% confidence intervals were calculated using raw values for the partial slopes and standard error, back-transforming the lower and upper bounds in the same manner.

## Results

We first conducted a statewide analysis including all counties in the state of Pennsylvania for an association between UNGD and asthma hospitalization admission rates (HAR). Because the statewide analysis showed strong urban and rural differences in asthma HAR, we additionally carried out a sensitivity analysis including only the rural counties. Both approaches showed an association between asthma and HAR.

**Table 1. Results of the multiple linear regression model using the 62 of the 67 counties in PA that report asthma HAR to investigate the relationship between UNGD and asthma hospitalization.**

| Model 1 | Range | Unit Increase | Associated % Change in Asthma HAR | 95% Confidence Intervals (as % Change) | p-value (α = 0.05) |
|---|---|---|---|---|---|
| **Annual well density** | 0.00–0.43 wells/sq mi | 0.01 wells/sq mi | +3.0% | [0.95%, 7.21%] | 0.000127 |
| **Median Income** | $34,018 - $86,093 | $1000 | -1.46% | [-1.83%, -1.10%] | $4.16 \times 10^{-15}$ |
| **% Urban Population** | 0–100% | 1% | +0.87% | [0.71%, 1.03%] | $< 2.0 \times 10^{-16}$ |
| **PM$_{2.5}$** | 7.8–21.5 μg/m$^3$ | 1 μg/m$^3$ | +2.70% | [0.25%, 5.23%] | 0.028029 |

Ranges for variables are based on counties included in the model. Since the response variable was natural log-transformed, the associated percent change in asthma HAR for each explanatory variable was computed from backtransforming the partial slopes of the regression model. The 95% confidence intervals were computed using the raw partial slope and standard error, and then backtransforming the lower and upper bounds. A percent change highlighted in green represents an associated percent increase; a percent change highlighted in red represents an associated percent decrease.

## Model 1: Multiple linear regression model for all counties

To investigate the relationship between asthma HAR and annual well density, we began by designing a multiple linear regression model for all counties in the state reporting asthma HAR while controlling for fixed-year effects. Our multiple linear regression model resulted in a significant relationship between asthma HAR and annual well density (Table 1). In this model, a 0.01 increase in annual well density is associated with a 3.0% increase in asthma HAR. Median income, percentage of population in urban areas, and PM$_{2.5}$ were all included as significant covariables. For median income, a $1000 increase on a county-level is associated with a 1.56% decrease in asthma HAR. The percentage of population in urban areas is positively associated with a 1% increase associated with a 0.87% increase in asthma HAR. Lastly, a 1 μg/m$^3$ increase in annual average concentration of PM$_{2.5}$ pollution is associated with an 2.70% increase in asthma HAR. This regression model has a prediction error rate of 16.07%, an adjusted R-squared value of 0.3211, and significant p-value (<0.05).

**Urban v. rural asthma hospitalization admission rates.** The risk of asthma hospitalization is well established for urban dwellers, so we compared asthma HAR in rural versus urban counties as a useful control for our study [29–32]. A line graph revealed the urban counties' average asthma HAR were consistently higher than the average asthma HAR of rural counties in every year from 2001 to 2014 (Fig 2B). Notably, there was more variability in the asthma HAR in urban counties due to inclusion of counties with historically high asthma HAR, such as Philadelphia. Beginning in about 2010, there is a distinct downward trend for average asthma HAR for both rural and urban counties over time. Rather than a two-sample t-test, a Wilcoxon sum rank test was performed due to the small number of data points being compared and that our data otherwise met the test's assumptions. This test resulted in a significant p-value of $1.31 \times 10^{-4}$, indicating there is a significant difference between the distribution of average asthma HAR for urban & rural counties between 2001–2014.

## Model 2: Multiple linear regression model for rural counties

To refine our analysis to consider only rural areas, we designed a multiple linear regression model to include only data for rural counties of Pennsylvania (Fig 3). After finalizing the model with fixed-year effects, the multiple linear regression analysis showed a significant relationship between asthma HAR and annual well density (Table 2). In this model, a 0.01 increase in annual well density is associated with a 3.31% increase in asthma HAR. Significant covariables in this model include median household income, percentage of white population, and

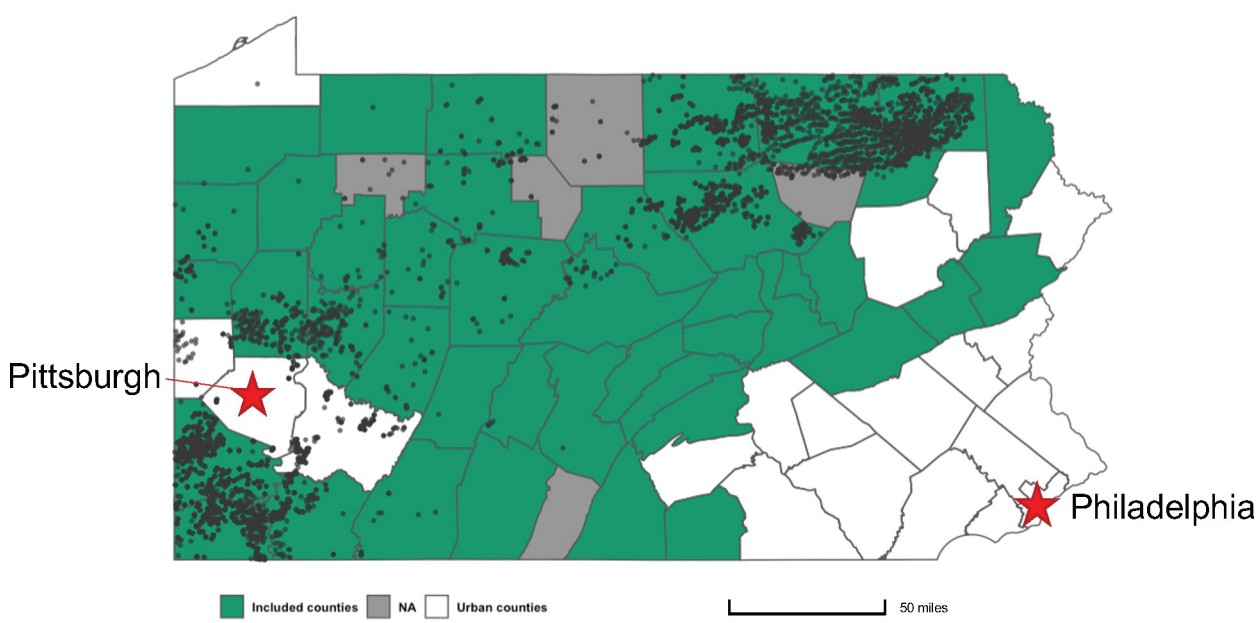

**Fig 3. Pennsylvania counties included for regression model optimization.** Active UNGD wells spudded through the end of 2014 shown by grey circles. The color designation of counties are as follows: green indicates rural counties that were included, grey indicates counties that lack or do not display asthma HAR for all years of interest, and white indicates urban counties that were excluded.

$PM_{2.5}$ pollution. For median household income, a $1000 increase on a county-level is associated with a 1.90% decrease in asthma HAR. We found that the percentage of white population is also negatively associated, with a 1.0% increase in the percentage of white population associated with a 2.54% decrease in asthma HAR. Lastly, a 1 µg/m$^3$ increase in annual average concentration of $PM_{2.5}$ pollution is associated with an 7.33% increase in asthma HAR. This regression model has a prediction error rate of 15.76%, an adjusted R-squared value of 0.2899, and significant p-value ($<0.05$).

**Using outdoor $PM_{2.5}$ as measurement of air quality.** Particulate matter is one of the six criteria air pollutants identified by the EPA to harm human health and commonly used as a proxy for overall air quality [33]. In our study, we relied on outdoor $PM_{2.5}$ as a general county-level metric for air pollution connected to combustion-based power generation, such as coal, and traffic emissions. Similar for asthma HAR, we observed a statewide downward trend for

**Table 2. Results of the multiple linear regression model using only rural counties to investigate the relationship between UNGD and asthma hospitalization.**

| Model 1 | Range | Unit Increase | Associated % Change in Asthma HAR | 95% Confidence Intervals (as % Change) | p-value ($\alpha = 0.05$) |
|---|---|---|---|---|---|
| **Annual well density** | 0.00–0.43 wells/sq mi | 0.01 wells/sq mi | +3.0% | [0.95%, 7.21%] | 0.000127 |
| **Median Income** | $34,018 - $86,093 | $1000 | -1.46% | [-1.83%, -1.10%] | $4.16 \times 10^{-15}$ |
| **% Urban Population** | 0–100% | 1% | +0.87% | [0.71%, 1.03%] | $< 2.0 \times 10^{-16}$ |
| **PM $_{2.5}$** | 7.8–21.5 µg/m$^3$ | 1 µg/m$^3$ | +2.70% | [0.25%, 5.23%] | 0.028029 |

Ranges for variables are based on counties included in the model. Since the response variable was natural log-transformed, the associated percent change in asthma HAR for each explanatory variable was computed from through backtransforming the partial slopes of the regression model. The 95% confidence intervals were computed using the raw partial slope and standard error, and then backtransforming the lower and upper bounds. A percent change highlighted in green represents an associated percent increase, while highlight in red represents an associated percent decrease.

annual average ambient concentrations of $PM_{2.5}$ as we did for asthma HAR over time in both urban and rural counties (S1 Fig). Although not statistically significant according to a Wilcoxon sum rank test, there was a consistent trend of urban counties having higher $PM_{2.5}$ pollution on average than rural counties over time. In a simple regression between asthma HAR and $PM_{2.5}$ pollution, there is a strong positive relationship, which was also observed in the statewide multiple linear regression analyses (S1 Table). Notably, this simple regression model was slightly improved with a 1-year lag on the annual average $PM_{2.5}$ level (S2 Table).

## Discussion

We tested two models in our analysis, Model 1: statewide for all counties. Because of the significant differences in air pollution and higher asthma HAR in urban counties, we developed Model 2: Rural alone, which compared only rural counties. Also, in Pennsylvania a majority of the wells are in rural counties. When we analyzed only rural counties across the state, we still observed a significant association between asthma HAR and annual well density after controlling for several confounding variables [34].

We found a significant association between asthma HAR and unconventional annual well density at the county-level in Pennsylvania when considering all counties, after controlling for several potentially confounding variables. The association was seen when we adjusted for the urban metric of counties and when we examined the rural counties alone. Our models were validated by finding several well-established associations with asthma. (1) We found a negative association between income and hospitalization as previously observed in studies on the influence of socioeconomic status on asthma [35]. (2) There was a strong positive relationship between the percent of individuals in a county living in urban areas and asthma hospitalization. This relationship may be representative of asthma triggers associated with urban living, such as pollution, violence, and housing quality [29–32]. (3) We also observed a positive association between annual $PM_{2.5}$ concentrations and asthma HAR [36]. Air pollution, specifically $PM_{2.5}$, is a well-known environmental trigger for asthma [11]. Although excluded from the model reported in this study due to issues of multicollinearity, we observed a strong racial disparity across the state, with the percentage of white population negatively associated with asthma hospitalization rates.

Previous studies have indicated that geographic proximity to UNGD is associated with adverse health outcomes, including increases in asthma [5, 7, 14, 37, 38]. Our general exposure metric of well density has a low spatial granularity compared to methods using precise residential geocoding from patient charts, such as zip codes or home addresses. By using well density, we are forced to assume a uniform well distribution over a county's land area. Despite this assumption, a clear association can still be observed at the county-level, likely due to the high UNGD and supporting activity during our study period. The populations of the highest UNGD counties, Bradford, Washington, Susquehanna, and Greene are exposed extensively as these counties all had a cumulative well density close to or over 1 unconventional well per square mile by the end of 2014. Other studies found that patients within about a mile were significantly affected [7, 39].

An unexpected finding was a significant reduction in average asthma HAR since 2010. The decrease was seen throughout the state when counties were separated in rural and urban groups. This decrease parallels the downward trend of asthma hospitalization rates throughout the nation for the same period [40]. Several changes in health care management could impact asthma HAR during our study period. The first was the passage of the Affordable Care Act in 2010 (Obamacare), and the subsequent expansion of Medicaid enrollment under this law in Pennsylvania in 2015. These changes provided health insurance to 1.1 million people in the

state [41]. The second was the increase in corticosteroid prescriptions [42]. Asthma has a chronic inflammatory component that can be managed by regular medication with inhaled corticosteroids, thus reducing the severe exacerbations that cause hospitalizations. The large number of newly insured patients likely improved the management of their asthma. One consequence of better management might be increased corticosteroid usage.

Hydraulic fracturing could also have contributed to steady declines in asthma HAR as the boom in natural gas extraction led to a reduction of the use of coal for electricity generation from 31% to 11% between 2006 and 2016 [43]. Coal powered electrical generation plants are a significant source of $PM_{2.5}$, which exacerbates asthma. Since 2011, Pennsylvania retired six coal powered generators in the following counties: Beaver (2), Chester, Snyder, Washington, and Greene. The latter three counties are designated as rural according to our definition based on population density. We note that the drop in asthma HAR was seen statewide and not confined to these counties, and that from the list of counties with retired coal generators, only Washington county is in the highest quartile of HF counties. Thus, while asthmatics might benefit from HF by replacing coal with cleaner natural gas as an energy source, the benefits may be reduced for asthmatics living near HF wells.

We also observed a strong racial disparity in asthma HAR (Table 2). The percentage of white population was negatively associated with asthma hospitalization. Therefore, the percentage of minority populations in rural counties (determined by non-white racial categories reported by the ACS, including Black, Native American, Asian, Pacific Islander, and biracial identities) is positively associated with asthma HAR, a health disparity for minorities that has been observed in past studies [44, 45]. Surprisingly, the disparity was seen even when the analysis was restricted to rural counties alone.

The last significant covariable for this model was annual $PM_{2.5}$ concentrations, which were positively associated with asthma HAR. For rural counties, $PM_{2.5}$ was associated with a percent increase in asthma HAR 3 times greater than when considering both urban and rural counties in the other statewide model. Outdoor $PM_{2.5}$ may be a more influential environmental trigger in rural areas relative to urban areas as rural patterns of activity and occupations may favor spending more time outdoors [46]. Although we did not observe a strong association between UNGD and $PM_{2.5}$ levels at the county-level in Pennsylvania, we have found that both UNGD and $PM_{2.5}$ were associated with asthma HAR separately (Table 1, Model 1). Other researchers have associated local $PM_{2.5}$ pollution with UNGD in Pennsylvania and other states [37, 47, 48]. However, we found that both UNGD and $PM_{2.5}$ were associated with asthma HAR but not each other indicating that they are each contributing to asthma HAR. We believe that association between the increased number of HF wells being spudded and the increase of asthmatics with severe exacerbations could be due to ineffective set-back distances.

This study illustrates the potential to address associative trends between UNGD and asthma exacerbation from publicly available data. The public data are less granular than data derived from protected health information, but typically have longer longitudinal frames as well as a wider geographical range. For example, this is the first statewide analysis of asthma and UNGD. Exploration of asthma exacerbation in other states with high UNGD, such as Colorado and Texas, using public data may also be informative, particularly if the other states have different asthma metrics publicly available that Pennsylvania lack, such as emergency department visitation.

### Limitations of the study

The major limitation of this study is that by solely relying on publicly sourced data, the data are significantly less granular than primary hospital records. The county level asthma HAR

permit only imprecise estimates of proximity to wells. Another limitation was inconsistent data availability for asthma HAR and relevant covariables. As of 2021, statewide public health data collection and management can be challenging as there are only six Pennsylvania counties and four municipalities with departments of public health. No publicly available data for asthma HAR was available for Forest, Cameron, Potter, Sullivan, and Fulton counties (S3 Table). Data was also sparse for 10 rural counties between 2005–2014, especially for Juniata, Montour, Snyder, and Union (S3 Table). Despite having asthma HAR dating to 2001, we were unable to consider data prior to 2005 due to a lack of reliable data for the majority of our covariables.

## Conclusion

Our study confirms and extends the growing literature associating UNGD with asthma HAR. Our models were robust enough to confirm an expected association of asthma HAR with urban areas, $PM_{2.5}$, income, and racial disparities (in model 2) using public data only. A major contribution of our work is that we utilized publicly available data without any protected health information. Both publicly available data and protected health information can contribute to our understanding of the health risks of environmental pollutants. Performing analyses using publicly available data enables the results to be shared widely and helps support research conducted on protected health information that is not publicly accessible. Future research investigating asthma exacerbation and UNGD should focus on elucidating casual mechanisms.

## Supporting information

**S1 Fig. Line graphs comparing trends of asthma HAR and average annual PM 2.5 concentrations in rural and urban counties.** Panel A shows a line graph for rural counties with average asthma HAR (per 10,000 cases) on the left axis in purple and average annual PM 2.5 concentrations ($\mu g/m^3$) on the right axis in gray over time. Panel B shows a line graph for urban counties of average asthma HAR (per 10,000 cases) on the left axis in purple and average annual PM 2.5 concentrations ($\mu g/m^3$) on the right axis in gray over time.
(PDF)

**S1 Table. Results from a model using rural PA counties: relationship between asthma HAR and average annual PM 2.5.** Ranges for PM 2.5 are based on counties included in the model. Since the response variable was natural log-transformed, the associated percent change in asthma HAR for the explanatory variable, PM 2.5, was computed from through backtransforming the partial slope. A percent change highlighted in green represents an associated percent increase.
(PDF)

**S2 Table. Results from a model using rural PA counties: relationship between asthma HAR and average annual PM 2.5 with a 1-year temporal lag.** Ranges for PM 2.5 are based on counties included in the model. Since the response variable was natural log-transformed, the associated percent change in asthma HAR for the explanatory variable, PM 2.5, was computed from through backtransforming the partial slope. A percent change highlighted in green represents an associated percent increase.
(PDF)

**S3 Table. List of the Pennsylvanian counties included in each model with information on age-adjusted asthma HAR data available and prevalence of HF wells.** Cells highlighted in blue indicate counties that did not have data available for age-adjusted asthma HAR for the

years of study. Cells highlighted in yellow indicate counties with incomplete data available for age-adjusted asthma HAR for the years of study. Cells with * indicates a county in the third quartile for HF well count based on the cumulative well count in 2014. Cells with ** indicates a county in the fourth quartile for HF well count based on the cumulative well count in 2014. (PDF)

## Acknowledgments

We thank, Nancy McKeon, Gabriela Daza, Siddhi Deshpande, and Dr. Meher Khan for helpful discussions. We thank the Short Term Educational Experiences for Research in Environmental Science (STEER) program for guidance.

## Author Contributions

**Conceptualization:** Anna Bushong, Thomas McKeon, Mary Regina Boland, Jeffrey Field.

**Data curation:** Anna Bushong, Thomas McKeon.

**Formal analysis:** Anna Bushong, Thomas McKeon.

**Methodology:** Anna Bushong, Thomas McKeon, Mary Regina Boland.

**Resources:** Jeffrey Field.

**Software:** Anna Bushong.

**Supervision:** Jeffrey Field.

**Validation:** Mary Regina Boland.

**Visualization:** Anna Bushong, Thomas McKeon.

**Writing – original draft:** Anna Bushong.

**Writing – review & editing:** Anna Bushong, Thomas McKeon, Mary Regina Boland, Jeffrey Field.

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
