## [Decision Letter · Decision Letter 0]

19 Jan 2022

PONE-D-21-35951Publicly available data reveals association between asthma hospitalizations and unconventional natural gas development in PennsylvaniaPLOS ONE

Dear Dr. Bushong,

Thank you for submitting your manuscript to PLOS ONE. After careful consideration, we feel that it has merit but does not fully meet PLOS ONE’s publication criteria as it currently stands. Therefore, we invite you to submit a revised version of the manuscript that addresses the points raised during the review process.

The manuscript has been evaluated by two very experienced referees. They raise a number of significant issues, that must be handled before the manuscript is suitable for decision.

Please, read the comments by the referees and respond to them, point-by-point.

However, I will especially stress two of them;

1. The part dealing with the domestic politics has to be deleted from the manuscript. This is a journal with a global readership.

2.  I also agree with one of the reviewers that Model 3 could be deleted.

We look forward to receiving your revised manuscript.

Kind regards,

Kjell Torén, MD, PhD

Academic Editor

PLOS ONE

Journal Requirements:

“This study was supported in part by the National Institute of Health R25ES021649 and P30ES013508.”

We note that you have provided additional information within the Funding Section that is not currently declared in your Funding Statement. Please note that funding information should not appear in other areas of your manuscript. We will only publish funding information present in the Funding Statement section of the online submission form.

“This study was supported in part by the National Institute of Health R25ES021649 and P30ES013508 to JF. The funders had no role in study design, data collection and analysis, decision to publish, or preparation of the manuscript.”

5.  We note that Figures 4 and 6  in your submission contain [map/satellite] images which may be copyrighted. All PLOS content is published under the Creative Commons Attribution License (CC BY 4.0), which means that the manuscript, images, and Supporting Information files will be freely available online, and any third party is permitted to access, download, copy, distribute, and use these materials in any way, even commercially, with proper attribution. For these reasons, we cannot publish previously copyrighted maps or satellite images created using proprietary data, such as Google software (Google Maps, Street View, and Earth). For more information, see our copyright guidelines: http://journals.plos.org/plosone/s/licenses-and-copyright.

   a. You may seek permission from the original copyright holder of Figures 4 and 6 to publish the content specifically under the CC BY 4.0 license.  

Reviewer's Responses to Questions

**Comments to the Author**

1. Is the manuscript technically sound, and do the data support the conclusions?

Reviewer #1: Yes

Reviewer #2: Yes

2. Has the statistical analysis been performed appropriately and rigorously? 

Reviewer #1: Yes

Reviewer #2: Yes

3. Have the authors made all data underlying the findings in their manuscript fully available?

Reviewer #1: Yes

Reviewer #2: Yes

4. Is the manuscript presented in an intelligible fashion and written in standard English?

Reviewer #1: No

Reviewer #2: Yes

5. Review Comments to the Author

Reviewer #1: The article reports on using data routinely collected in the US state of Pennsylvania to investigate the association of hospital admission rates for asthma with unconventional natural gas production. Several changes are need before the manuscript would be ready for publication.

-Introduction

The Introduction is unnecessarily long at approximately 1,000 words. Several paragraphs address the EPA’s failed attempt to impose the “secret science rule” and the advantage of using publicly available data. The text addressing these issues is excessively long, and the same could be accomplished in about half the number of words. Since the EPA never implemented this rule, it is unclear this is a particularly strong motivation, especially since it is likely other studies that rely on personal health information provide more convincing evidence.

Most of the last paragraph of the Introduction (except the final sentence) belongs in the Methods or the Results. This paragraph should clearly state the goals and objectives of the study.

-Methods

Minor comment: Explain the meaning of “spud” when first used in the text, which is in the second paragraph of the Methods.

The model building part of the Methods is unclear. Please clarify which variables were candidates for inclusion in the different full models, whether the models were built using forward addition or backward elimination or some combination of the two, and the criteria for inclusion or exclusion of candidate variables. The criteria for including variables that are potential confounders should not be an alpha of 0.05. Statistical significance alone is insufficient to judge potential confounding because variables with p values greater than 0.05 can still confound the effect of the exposure of interest. Use a higher p value (e.g., 0.10) as the criterion for which variables are included/retained as potential confounders. Also, the point of the models is to estimate the effect of Annual Well Density, so this variable should never be deleted from the model, as the authors did in the 3rd model.

-Results

The first paragraph of the Results belongs in the Methods.

In the second paragraph of the Results, the authors describe results from Model 1 in the following way: “In this model, a 0.01 increase in annual well density correlates to a 3.0% increase in asthma HAR.” The words “correlates to” are not optimal in this setting, partly because the results are not from a test of correlation. I expected to see something like the following, in which results are presented as the association of the health outcome with the exposure of interest: “In this model, a 3.0% increase in asthma HAR was associated with a 0.01 increase in annual well density.” The “correlates to” and similar expressions should be changed throughout the manuscript.

Figure 3 includes results from Model 1 presented in both a table and a graph. In the table, delete “Correlated” from the column heading “Correlated % Change in Asthma HAR.” In this column, put a negative sign before any result that is a percentage decrease to improve clarity of communication. Coloring the table cell is insufficient to indicate the direction of the effect. Follow this column with a new column for 95% confidence intervals for the effect estimates. The table alone communicates all relevant information, so the graph is redundant and should be deleted. These comments apply to Figures 5 and 7 as well. After deleting the graphs, re-label these three figures as tables.

Section on Model 2: The first sentence of this section is an example of using more words than needed. The original sentence is: “To refine our analysis to consider only rural areas, we designed a multiple linear regression model intended to optimize the study design for the publicly available data to include all available data for rural counties of Pennsylvania (Figure 4).” I suggest the following instead: “To refine our analysis to consider only rural areas, we designed a multiple linear regression model to include _only_ data for _the_ rural counties of Pennsylvania (Figure 4).” Sentences like this appear throughout the manuscript and are candidates for similar modification.

-Discussion

The authors did a good job with the Discussion, providing context for the findings and identifying limitations.

I recommend commenting on the potential for using the publicly available data to conduct ongoing surveillance. It appears they could be used to monitor longitudinal trends, which might exceed what most studies based on personal health information can accomplish.

Reviewer #2: GENERAL COMMENTS

This well-written manuscript reports the results of an ecological study of the association of area-density of unconventional oil and gas extraction (hydraulic “fracking”) and asthma hospitalizations in the U.S. state of Pennsylvania. The study methods and data analysis are appropriate if a bit conventional. The results of the study are not novel, but the contribution of the study lies in the confirmatory support this state-wide analysis brings to the fracking-asthma association observed in previous smaller studies. The county-wide frame used by the authors in this ecological analysis is characterized by exposure misclassification due to lack of individual patient residential address, and the lack of individual covariate data may mean that there is residual confounding of the observed association. That said, the exposure misclassification likely biases towards the null and the covariate adjustment is reasonable given the county-level analysis frame.

The authors state that the primary rationale for their study was to use publicly available data so the results could be used by the U.S. EPA for regulatory purposes given the Trump Administration’s efforts to promulgate a “Secret Science” rule. While this is a laudable rationale, the rule was blocked by court action and subsequently rescinded by the Biden EPA in early 2021. To this reviewer, the manuscript could be both shortened and better focused if the text on the Trump EPA “Secret Science” rule was eliminated from the manuscript, including the Abstract.

The authors attempt to replicate the results of a previous Johns Hopkins study using Geisinger Clinic patient data – Model 3. I also suggest dropping this component of the manuscript. It is unnecessary and distracting. The results of the state-wide analysis stand on their own.

SPECIFIC COMMENTS

The lack of page numbers makes the review unnecessarily difficult.

Introduction, 2nd page, 2nd full paragraph The statement “This rule was revoked when a minor procedural flaw in its final implementation was challenged in court” Is both factually incorrect – not a minor procedural flaw (see https://eelp.law.harvard.edu/2021/03/final-secret-science-rule/ -- and out of date. As noted above, I suggest eliminating all mention of this now rescinded rule in the manuscript. It is unnecessary and distracting.

Methods, Unconventional Oil & Gas Wells by County section “Spud” is a technical jargon term that should be explained at first use in the manuscript.

Discussion, 2nd page, 2nd full paragraph “Cortical steroid” should be replaced by “corticosteroid.”

6. PLOS authors have the option to publish the peer review history of their article (what does this mean?). If published, this will include your full peer review and any attached files.

Reviewer #1: No

Reviewer #2: **Yes: **John R. Balmes, MD

---

## [Author Response · Author response to Decision Letter 0]

26 Feb 2022

Response to Reviewers:

Date: Jan 19 2022 10:21AM

To: "Jeffrey M. Field" jfield@mail.med.upenn.edu

From: "PLOS ONE" plosone@plos.org

Subject: PLOS ONE Decision: Revision required [PONE-D-21-35951]

PONE-D-21-35951

Publicly available data reveals association between asthma hospitalizations and unconventional natural gas development in Pennsylvania

PLOS ONE

Dear Dr. Bushong,

Thank you for submitting your manuscript to PLOS ONE. After careful consideration, we feel that it has merit but does not fully meet PLOS ONE’s publication criteria as it currently stands. Therefore, we invite you to submit a revised version of the manuscript that addresses the points raised during the review process.

The manuscript has been evaluated by two very experienced referees. They raise a number of significant issues, that must be handled before the manuscript is suitable for decision.

Please, read the comments by the referees and respond to them, point-by-point.

However, I will especially stress two of them.

1. The part dealing with the domestic politics has to be deleted from the manuscript. This is a journal with a global readership.

Author’s Response: We removed discussion of politics. 

2. I also agree with one of the reviewers that Model 3 could be deleted.

Author’s Response: We removed Model 3. 

We look forward to receiving your revised manuscript.

Kind regards,

Kjell Torén, MD, PhD

Academic Editor

PLOS ONE

Journal Requirements:

Author’s Response: Thank you to the academic editor for this clarification. We reformatted the manuscript to fit PLOS ONE’s style requirements. Minor edits and edits for style were not tracked in the “track changes draft” so the reviewers can focus on the responses to their points.

“This study was supported in part by the National Institute of Health R25ES021649 and P30ES013508.”

We note that you have provided additional information within the Funding Section that is not currently declared in your Funding Statement. Please note that funding information should not appear in other areas of your manuscript. We will only publish funding information present in the Funding Statement section of the online submission form.

“This study was supported in part by the National Institute of Health R25ES021649 and P30ES013508 to JF. The funders had no role in study design, data collection and analysis, decision to publish, or preparation of the manuscript.”

Author’s Response: This is the appropriate funding statement, which we also stated in the cover letter for the revisions. “This study was supported in part by the National Institute of Health R25ES021649 and P30ES013508 to JF. The funders had no role in study design, data collection and analysis, decision to publish, or preparation of the manuscript.” 

Author’s Response: We included a citation for this statement in the revised manuscript.

Author’s Response: Thank you to the editor for clarifying this requirement for the Supporting Information. Captions for the Supporting Information files have been added at the end of the manuscript and in-text citations changed to match this formatting.

5. We note that Figures 4 and 6 in your submission contain [map/satellite] images which may be copyrighted. All PLOS content is published under the Creative Commons Attribution License (CC BY 4.0), which means that the manuscript, images, and Supporting Information files will be freely available online, and any third party is permitted to access, download, copy, distribute, and use these materials in any way, even commercially, with proper attribution. For these reasons, we cannot publish previously copyrighted maps or satellite images created using proprietary data, such as Google software (Google Maps, Street View, and Earth). For more information, see our copyright guidelines: http://journals.plos.org/plosone/s/licenses-and-copyright.

 a. You may seek permission from the original copyright holder of Figures 4 and 6 to publish the content specifically under the CC BY 4.0 license. 

Author’s Response: Both figures were created using publicly available data sourced directly from the Pennsylvania Department of Environmental Protection for the well locations and the spatial data for county boundaries from the Pennsylvania Department of Transportation, which was accessed through an open geospatial data portal known as PASDA. We cited these data sources initially but pending acceptance we will upload the primary data and reproducible code to GitHub. The files will be for Figure 4 (now revised in-text as Figure 3). Figure 6 was removed during the process of removing Model 3 from the manuscript. 

Reviewer's Responses to Questions

Comments to the Author

1. Is the manuscript technically sound, and do the data support the conclusions?

Reviewer #1: Yes

Reviewer #2: Yes

2. Has the statistical analysis been performed appropriately and rigorously? 

Reviewer #1: Yes

Reviewer #2: Yes

3. Have the authors made all data underlying the findings in their manuscript fully available?

Reviewer #1: Yes

Reviewer #2: Yes

4. Is the manuscript presented in an intelligible fashion and written in standard English?

Reviewer #1: No

Reviewer #2: Yes

5. Review Comments to the Author

Reviewer #1 

Reviewer #1: The article reports on using data routinely collected in the US state of Pennsylvania to investigate the association of hospital admission rates for asthma with unconventional natural gas production. Several changes are need before the manuscript would be ready for publication.

-Introduction

The Introduction is unnecessarily long at approximately 1,000 words. Several paragraphs address the EPA’s failed attempt to impose the “secret science rule” and the advantage of using publicly available data. The text addressing these issues is excessively long, and the same could be accomplished in about half the number of words. Since the EPA never implemented this rule, it is unclear this is a particularly strong motivation, especially since it is likely other studies that rely on personal health information provide more convincing evidence.

Author’s Response: Thank you for this comment. Our motivation to initiate the study was initially sparked by the “secret science rule”, but defer to the judgment of the reviewer and editor that discussion of this rule detracts from the manuscript and the core findings. All mention of the EPA’s rescinded rule has been removed from the introduction and discussion, though we continue to emphasize the public sourcing of the data, without the use of protected health information, as well as discuss the limitations of public datasets. 

Most of the last paragraph of the Introduction (except the final sentence) belongs in the Methods or the Results. This paragraph should clearly state the goals and objectives of the study.

Author’s Response: Thank you for this helpful comment. We have moved this portion of the introduction to the beginning of the Results and refocused the last paragraph of the introduction to clearly define the main goal and objective of the study. 

-Methods

Minor comment: Explain the meaning of “spud” when first used in the text, which is in the second paragraph of the Methods.

Author’s Response: Thank you for this helpful comment. In response, we added the definition of “spud” according to the Pennsylvania Department of Environmental Protection when the term is first used in the legend of Figure 1 and the first time it was used in the body of the manuscript. 

The model building part of the Methods is unclear. Please clarify which variables were candidates for inclusion in the different full models, whether the models were built using forward addition or backward elimination or some combination of the two, and the criteria for inclusion or exclusion of candidate variables. The criteria for including variables that are potential confounders should not be an alpha of 0.05. Statistical significance alone is insufficient to judge potential confounding because variables with p values greater than 0.05 can still confound the effect of the exposure of interest. Use a higher p value (e.g., 0.10) as the criterion for which variables are included/retained as potential confounders. Also, the point of the models is to estimate the effect of Annual Well Density, so this variable should never be deleted from the model, as the authors did in the 3rd model.

Author’s Response: We thank the reviewer for this comment. We have clarified this section in the methods to help clarify our system of selecting variable and assessment of each variables' inclusion in the model along with collinearity assessment to create more parsimonious models. Furthermore, one of the models - was designed to most closely fit the prior published literature (Model 3). Though model 3 is no longer in the manuscript, it is a useful comparator point for variables accepted in the literature, so we included all terms that they included and cite the study as well as two other studies (refs 7, 14 and 28). 

-Results

The first paragraph of the Results belongs in the Methods.

Author’s Response: The authors agree this paragraph is more appropriate under the Methods since it explains reasoning and sourcing of publicly available data. It has been moved to be the first paragraph of the methods section. 

In the second paragraph of the Results, the authors describe results from Model 1 in the following way: “In this model, a 0.01 increase in annual well density correlates to a 3.0% increase in asthma HAR.” The words “correlates to” are not optimal in this setting, partly because the results are not from a test of correlation. I expected to see something like the following, in which results are presented as the association of the health outcome with the exposure of interest: “In this model, a 3.0% increase in asthma HAR was associated with a 0.01 increase in annual well density.” The “correlates to” and similar expressions should be changed throughout the manuscript.

Author’s Response: Thank you to the reviewer for this important comment. You are correct with the suboptimal usage of “correlated”, rather than the appropriate term “associated”. This has been changed through the body of manuscript and supplementary materials, including the relevant figures and tables. 

Figure 3 includes results from Model 1 presented in both a table and a graph. In the table, delete “Correlated” from the column heading “Correlated % Change in Asthma HAR.” In this column, put a negative sign before any result that is a percentage decrease to improve clarity of communication. Coloring the table cell is insufficient to indicate the direction of the effect. Follow this column with a new column for 95% confidence intervals for the effect estimates. The table alone communicates all relevant information, so the graph is redundant and should be deleted. These comments apply to Figures 5 and 7 as well. After deleting the graphs, re-label these three figures as tables.

Author’s Response: Thank you for this comment. We have made the suggested improvements to these tables and agree these changes have improved the presentation of the results. The manuscript has been edited to reflect the re-labeling of these figures as tables. A new column for 95% confidence intervals for the effect estimates was also created, reporting these intervals as percent change. The graphs were also removed as suggested. 

Section on Model 2: The first sentence of this section is an example of using more words than needed. The original sentence is: “To refine our analysis to consider only rural areas, we designed a multiple linear regression model intended to optimize the study design for the publicly available data to include all available data for rural counties of Pennsylvania (Figure 4).” I suggest the following instead: “To refine our analysis to consider only rural areas, we designed a multiple linear regression model to include _only_ data for _the_ rural counties of Pennsylvania (Figure 4).” Sentences like this appear throughout the manuscript and are candidates for similar modification.

Author’s Response: Thank you for this helpful comment. We corrected this sentence as suggested and have carefully reviewed the manuscript making appropriate changes to sentences when found. Note that not all of the rephrasing was saved in the track changes draft for clarity of reading.

-Discussion

The authors did a good job with the Discussion, providing context for the findings and identifying limitations.

I recommend commenting on the potential for using the publicly available data to conduct ongoing surveillance. It appears they could be used to monitor longitudinal trends, which might exceed what most studies based on personal health information can accomplish.

Author’s Response: Thank you to the reviewer for this comment. We agree explicitly commenting on this benefit of publicly available data improves the discussion. Therefore, we incorporated a statement regarding the potential for ongoing surveillance in the last paragraph of the Discussion. 

Reviewer #2 

This well-written manuscript reports the results of an ecological study of the association of area-density of unconventional oil and gas extraction (hydraulic “fracking”) and asthma hospitalizations in the U.S. state of Pennsylvania. The study methods and data analysis are appropriate if a bit conventional. The results of the study are not novel, but the contribution of the study lies in the confirmatory support this state-wide analysis brings to the fracking-asthma association observed in previous smaller studies. The county-wide frame used by the authors in this ecological analysis is characterized by exposure misclassification due to lack of individual patient residential address, and the lack of individual covariate data may mean that there is residual confounding of the observed association. That said, the exposure misclassification likely biases towards the null and the covariate adjustment is reasonable given the county-level analysis frame.

The authors state that the primary rationale for their study was to use publicly available data so the results could be used by the U.S. EPA for regulatory purposes given the Trump Administration’s efforts to promulgate a “Secret Science” rule. While this is a laudable rationale, the rule was blocked by court action and subsequently rescinded by the Biden EPA in early 2021. To this reviewer, the manuscript could be both shortened and better focused if the text on the Trump EPA “Secret Science” rule was eliminated from the manuscript, including the Abstract.

Author’s Response: We are encouraged by the reviewer’s positive comments. We defer to the reviewer’s comments that focus on the “Secret Science” rule distracts from the focus of the manuscript as mentioned in response to the first reviewer and have eliminated it from the manuscript.

The authors attempt to replicate the results of a previous Johns Hopkins study using Geisinger Clinic patient data – Model 3. I also suggest dropping this component of the manuscript. It is unnecessary and distracting. The results of the state-wide analysis stand on their own.

Author’s Response: Thank you for this straightforward, helpful comment. The authors agreed the manuscript would benefit from streamlining and have removed model 3. The remaining portion of the manuscript has been edited to reflect this change. 

SPECIFIC COMMENTS

The lack of page numbers makes the review unnecessarily difficult.

Introduction, 2nd page, 2nd full paragraph The statement “This rule was revoked when a minor procedural flaw in its final implementation was challenged in court” Is both factually incorrect – not a minor procedural flaw (see https://eelp.law.harvard.edu/2021/03/final-secret-science-rule/ -- and out of date. As noted above, I suggest eliminating all mention of this now rescinded rule in the manuscript. It is unnecessary and distracting.

Author’s Response: We added page numbers. As discussed above, we eliminated all mention of the rescinded rule from the manuscript. 

Methods, Unconventional Oil & Gas Wells by County section “Spud” is a technical jargon term that should be explained at first use in the manuscript.

Author’s Response: Thank you to the reviewer for this comment. “Spud” is now defined when first used in the legend of “Figure 1” and in the body of the manuscript to clarify the meaning of this technical term. 

Discussion, 2nd page, 2nd full paragraph “Cortical steroid” should be replaced by “corticosteroid.”

Author’s Response: Thank you for this comment. “Cortical steroid” has been replaced by “corticosteroid” throughout the manuscript.

---

## [Editor Report · Decision Letter 1]

3 Mar 2022

Publicly available data reveals association between asthma hospitalizations and unconventional natural gas development in Pennsylvania

PONE-D-21-35951R1

Dear Dr. Field,

We’re pleased to inform you that your manuscript has been judged scientifically suitable for publication and will be formally accepted for publication once it meets all outstanding technical requirements.

Kind regards,

Kjell Torén, MD, PhD

Academic Editor

PLOS ONE

Additional Editor Comments (optional):

The manuscript has been considerably improved, and I consider that it is ready for acceptance, i.a. the manuscript is accepted.
---

## [Editor Report · Acceptance letter]

22 Mar 2022

PONE-D-21-35951R1 

Publicly available data reveals association between asthma hospitalizations and unconventional natural gas development in Pennsylvania 

Dear Dr. Field:

I'm pleased to inform you that your manuscript has been deemed suitable for publication in PLOS ONE. Congratulations! Your manuscript is now with our production department. 

Kind regards, 

on behalf of

Dr. Kjell Torén 

Academic Editor

PLOS ONE